# Use of Deep Learning to Detect the Maternal Heart Rate and False Signals on Fetal Heart Rate Recordings

**DOI:** 10.3390/bios12090691

**Published:** 2022-08-27

**Authors:** Samuel Boudet, Agathe Houzé de l’Aulnoit, Laurent Peyrodie, Romain Demailly, Denis Houzé de l’Aulnoit

**Affiliations:** 1Faculty of Medicine and Midwifery, ETHICS EA 7446 Lille Catholic University, F-59000 Lille, France; 2Obstetrics Department, Lille Catholic Hospital, Lille Catholic University, F-59020 Lille, France; 3Junia Haut de France, F-59000 Lille, France

**Keywords:** fetal heart rate, maternal heart rate, cardiotocogram, gated recurrent unit, deep learning

## Abstract

We have developed deep learning models for automatic identification of the maternal heart rate (MHR) and, more generally, false signals (FSs) on fetal heart rate (FHR) recordings. The models can be used to preprocess FHR data prior to automated analysis or as a clinical alert system to assist the practitioner. Three models were developed and used to detect (i) FSs on the MHR channel (the FSMHR model), (ii) the MHR and FSs on the Doppler FHR sensor (the FSDop model), and (iii) FSs on the scalp ECG channel (the FSScalp model). The FSDop model was the most useful because FSs are far more frequent on the Doppler FHR channel. All three models were based on a multilayer, symmetric, GRU, and were trained on data recorded during the first and second stages of delivery. The FSMHR and FSDop models were also trained on antepartum recordings. The training dataset contained 1030 expert-annotated periods (mean duration: 36 min) from 635 recordings. In an initial evaluation of routine clinical practice, 30 fully annotated recordings for each sensor type (mean duration: 5 h for MHR and Doppler sensors, and 3 h for the scalp ECG sensor) were analyzed. The sensitivity, positive predictive value (PPV) and accuracy were respectively 62.20%, 87.1% and 99.90% for the FSMHR model, 93.1%, 95.6% and 99.68% for the FSDop model, and 44.6%, 87.2% and 99.93% for the FSScalp model. We built a second test dataset with a more solid ground truth by selecting 45 periods (lasting 20 min, on average) on which the Doppler FHR and scalp ECG signals were recorded simultaneously. Using scalp ECG data, the experts estimated the true FHR value more reliably and thus annotated the Doppler FHR channel more precisely. The models achieved a sensitivity of 53.3%, a PPV of 62.4%, and an accuracy of 97.29%. In comparison, two experts (blinded to the scalp ECG data) respectively achieved a sensitivity of 15.7%, a PPV of 74.3%, and an accuracy of 96.91% and a sensitivity of 60.7%, a PPV of 83.5% and an accuracy of 98.24%. Hence, the models performed at expert level (better than one expert and worse than the other), although a well-trained expert with good knowledge of FSs could probably do better in some cases. The models and datasets have been included in the Fetal Heart Rate Morphological Analysis open-source MATLAB toolbox and can be used freely for research purposes.

## 1. Introduction

The fetal heart rate (FHR) is a key parameter for monitoring fetal well-being during pregnancy, labor, and delivery. Accurate interpretation of the FHR is important for avoiding unnecessary cesarean sections and instrumental deliveries and reducing the risk of fetal acidosis. In France, the FHR is recorded during delivery (in all cases) and before delivery (for at-risk pregnancies only).

The FHR signal is analyzed by midwives and obstetricians for abnormalities such as decelerations, low variability, bradycardia, tachycardia, and sinusoidal patterns. The International Federation of Gynecology Obstetrics (FIGO) has issued guidelines on FHR analysis [1].

The FHR is measured with a Doppler sensor or a scalp electrocardiogram (ECG) sensor. The Doppler measurement is noninvasive, whereas the ECG sensor requires an incision on the fetus’ scalp: this is associated with a risk (albeit very low) for the fetus and often discomfort for the mother. Moreover, a Doppler sensor can be used at any time during pregnancy, whereas a scalp ECG electrode can only be used after rupture of the membranes and often falls off the scalp during the second stage of delivery. However, the Doppler sensor is less accurate—particularly when the FHR is highly variable—and is more subject to missing signals (MS) and false signals (FSs). This is why a Doppler sensor is used first and then replaced by a scalp ECG sensor when the signal is ambiguous (i.e., potentially containing an FS), when MSs are present, or when FHR variability must be measured accurately [2]. In our maternity hospital, a scalp ECG electrode is used in around 10% of deliveries; obviously, this proportion varies from one institution to another. The FHR can be also measured with abdominal ECG sensors, although this technique is still not widespread and falls outside the scope of the present study [3,4].

Using the raw data from the Doppler sensor or the scalp ECG sensor, the FHR is calculated on the cardiotocograph (CTG) monitor by applying a proprietary algorithm based on auto-correlation. The details of these proprietary algorithms have not been published by the manufacturers. The algorithms can often output false values that correspond to a harmonic of the true FHR value: double the rate, half the rate, or (more rarely) triple the rate. Moreover, the Doppler FHR sensor often records the maternal heart rate (MHR) or a harmonic of the latter, rather than the FHR; the signal is then considered to be false. Lastly, noisy raw data can give rise to random values – albeit only for a few seconds. All these signals will be referred to here as FSs. Although scalp ECG sensors can reportedly sometimes measure the MHR (particularly in cases of fetal death) [5], we have not found any examples in our dataset. FSs on scalp ECGs mainly correspond to rare, short periods of random values.

In most cases, an expert can easily identify FSs. However, this process might be critical because some FSs look like pathologic FHR signals: for example, a switch from an FHR value to an MHR value may look like a deceleration or bradycardia in the fetus. Moreover, maternal tachycardia can produce a signal that looks like the FHR, and so the true FHR might not be analyzed for an hour. Misinterpretation of the MHR as the FHR is relatively common. According to Reinhard et al. [6], MHR periods are found in up to 90% of intrapartum recordings and account for 6.2% of the duration of the recording. This problem is particularly frequent during the second stage of delivery and can be particularly dangerous for the fetus [7,8]. For example, France’s Melchior classification (used to describe the second stage of delivery [9]) suggests an erroneous type 3 FHR pattern (corresponding to bradycardia plus accelerations synchronized with uterine contractions (UCs), and for which expulsive efforts should last for less than 15 min) these cases (accounting for ≈4% of deliveries) correspond in fact to MHR interferences [10,11]. This error highlights the extent of the problem.

To help identify FSs, modern cardiotocographs are equipped with an MHR sensor: this is either an ECG sensor combined with the tocometer sensor on the belt (referred to as the “TOCO+”), or an oximeter on the mother’s finger. Superposition of the MHR and the FHR suggests that the latter is an FHR FS, although natural coincidences can occur. Moreover, the MHR channel has often periods of MS and FS (generally harmonics that are double, triple or half the true rate), particularly during the second stage of delivery, thus complicating the analysis.

To avoid ambiguity, we defined signal loss or an MS as a period during which the CTG device did not send MHR or FHR values. Furthermore, we defined an FS as any measured signal that did not correspond to the sensor’s target heart rate (i.e., periods of MHR recorded by a Doppler FHR sensor, signal harmonics, or other aberrant values). We did not consider that an inaccurate heart rate was an FS, even though this distinction is not always obvious.

The practitioner can use several signs to differentiate between FSs and true signals (TSs), as summarized in Appendix A. Many practitioners are apparently unaware of some of these signs, and so their ability to differentiate between FSs and TSs could probably be improved.

FSs also impede the automatic analysis of FHR recordings. Several automatic methods for FHR analysis have been developed in the last few years, notably with a view to preventing acidosis during delivery. Most of these methods include a preprocessing step in which FSs are partially removed [12]. This generally consists in detecting short periods during which the measured rate is significantly higher or lower (by more than 25 bpm, typically) than in the preceding period. We are not aware of any publications on the accurate identification of long periods of MHR, particularly when part of the MHR signal is missing. The closest reference to this problem was made by Pinto et al. [13], who considered that there are MHR-FHR ambiguities when the difference between MHR and FHR is less than 5 bpm; accordingly, the researchers removed these periods before further analysis. However, this method cannot be applied when the MHR is missing, when the FHR almost coincides with MHR, or when the FS is a harmonic of the FHR or the MHR. Pinto et al.’s results nevertheless emphasized that periods of MHR recording introduce significant bias into automated FHR feature detection.

The software in CTG monitors sometimes also comprises an alert system based on coincidence between the MHR and FHR channels [8]. However, these coincidences have to be checked manually. Moreover, the alert is not triggered if the MHR sensor does not record a signal during this period. This problem is often neglected in the literature [14]) because it merely constitutes a preprocessing step. However, FSs are known to constitute a major source of error in FHR analysis, whether visual [15,16] or automatic [13].

Deep learning (DL) models have recently emerged in which the FHR is used directly as an input [17,18,19]. There are no prior feature extraction steps, and features can emerge automatically from the models’ architectures and the clinical outcome (often the arterial umbilical cord blood pH) used as the output. Thus, one could imagine that the concept of an FS could emerge from a DL model. However, given (i) the relatively weak link between the FHR and the clinical outcome, and (ii) the complexity of the problem, we suspected that even several tens of thousands or hundreds of thousands of delivery recordings might not be enough to prompt the emergence of such complex features. We therefore sought to design models for these specific tasks. In particular, we sought to determine a solid ground truth for FSs (i.e., better than an expert could produce using the same information as the model) and thus increase the models’ effectiveness.

The objectives of the present study were to develop the first intelligent methods for the automatic recognition of FSs and to present this problem to other signal processing researchers. We developed three models for the detection of FSs on CTG recordings. Firstly, the FSMHR model detected FS periods on the MHR channel (either from ECG sensor on the tocometer or from the finger oximeter). Secondly, the FSDop model detected MHR and FS periods on the Doppler FHR channel. Thirdly, the FSScalp model detected FS periods on the scalp ECG channel. All three models have been established for the first and second stages of delivery, and the FSMHR and FSDop models have also been established for antepartum recordings. FSDop is the most complicated and clinically important model, given the several possible sources of FSs (harmonics, the MHR, and other aberrant signals).

The work described below was based on recordings from Philips CTG monitors. Although our methods should apply to other brands, some of the preprocessing steps are probably brand-specific and would have to be adjusted for use with other monitors.

Application of these models might help to (i) improve automatic FHR analysis methods (particularly for acidosis prediction during delivery), (ii) develop a smart alert system that tells the practitioner to reposition the FHR sensor or replace it with a scalp ECG electrode, and (iii) indicate when the FHR sensor is in fact measuring the MHR (thus avoiding potentially dangerous misinterpretations for the fetus).

Below, we describe the models, their training, the two evaluation methods, and the results.

## 2. Description of the Models

### 2.1. Data Acquisition

The three models were trained and evaluated on data recorded at the Saint Vincent de Paul Maternity Hospital (Lille, France) and stored in the “Bien Naître” data warehouse (registered with the French National Data Protection Commission; reference: REG 077)). At the time of writing, this data warehouse contained 22,000 delivery recordings (recorded from 2011 onwards) and 5000 antepartum recordings (recorded from 2019 onwards) [20]. The data warehouse’s research objectives and procedures had been approved by the local institutional review board (CIER GHICL, Lille, France) on 4 August 2016 (reference: 2016-06-08). Each woman was informed about the inclusion of her newborn’s data in the data warehouse and gave her written consent to the storage and use of these data.

The CTG monitors (Avalon FM30 and FM20^®^, Philips Medical Systems, Best, The Netherlands) sent signals to a central, dedicated research server via an Ethernet or Wi-Fi connection, using an in-house solution [21].

All heart rate signals were acquired at a frequency of 4 Hz and a resolution of 0.25 bpm (or 1 bpm, for MHRs recorded with an oximeter). The tocometer signal was recorded at a frequency of 4 Hz and a resolution of 0.5 mmHg. MHR signals have only been recorded in our maternity hospital since April 2015; given the importance of the MHR for detecting FS, we did not include data recorded before this date.

### 2.2. Selection of the Training Dataset

Recordings were extracted from the “Bien Naître” data warehouse and used to train and validate the models. These recordings were also annotated by experts, constituting the ground truth. The training/validation dataset was composed of periods from 635 recordings (dating from 1 April 2015, to 31 December 2019) selected as follows:AA total of 94 perpartum recordings, corresponding to all the recordings containing periods of at least 10 min during which signals from scalp ECG and Doppler sensors were recorded simultaneously. These recordings were used to train FSDop, FSMHR and FSScalp. In the event of doubt, solid ground truth can be determined by experts, using the signal from the other sensor as an indicator (i.e., the scalp sensor for FSDop and the Doppler sensor for FSScalp). These recordings are also of value for the current problem because, if practitioners have positioned the two sensors, it is probably because the recordings show FS ambiguities.BA total of 38 antepartum recordings presenting marked variations (either FHR/MHR switches or decelerations), for training the FSDop and FSMHR models.CA total of 96 routine perpartum recordings annotated by the practitioner as having major FSs and MSs (for training the FSDop, FSScalp and FSMHR models).DA total of 107 perpartum recordings (all recorded in 2016) with data from the scalp ECG electrode but (in contrast to dataset A) lacking simultaneously recorded Doppler data. Nonetheless, the first part of these recordings was usually composed of the Doppler signal (often with MSs or FS ambiguities) and therefore was also analyzed. Thus, this dataset was used to train the FSDop, FSScalp and FSMHR models.EA ttoal of 300 perpartum recordings with a Doppler signal, selected for their utility by experts from among the 915 recordings made in 2016 but not already included in datasets A to D (for training the FSDop and FSMHR models).

On each recording, the experts selected at least one period ranging from 5 min to 4 h in duration. These periods were used to train the FSDop and/or FSMHR models on one hand or the FSScalp model on the other. Within these selected periods, the experts annotated segments with clearly TSs and segments with clearly FSs. A lack of annotation meant that either the experts were unsure or the period did not contain difficult-to-interpret features and so was not worth annotating.

In some cases, a high proportion of the selected periods was not annotated. The selected period was long enough to include all the information required for interpretation. For example, we assumed that MHR interference was captured by the Doppler FHR sensor for one minute. During this minute and the preceding 30 min, the MHR sensor was not in place. The selected period had to include all the signals starting a few minutes before the MHR sensor was removed, so that the model could estimate the range of possible MHRs during the annotated period.

On datasets (A), (B), and (C), the selected periods were fully annotated by experts blinded to the models’ outputs. Datasets (D) and (E) were annotated after the initial models had been trained, and so only parts that appeared to be more difficult to interpret or where the models were wrong (or not confident enough) were annotated. Thus, the dataset grew progressively as the models’ complexity increased.

Recordings were randomly attributed to the training dataset (80% of the total duration) or the validation dataset (20%). This attribution was performed for the FSDop/FSMHR datasets together and then for the FSScalp datasets separately. The validation dataset was used to stop the training early (to avoid overfitting) and to compare the various models and hyperparameters. The final evaluation was carried out with the test datasets described in Section 3.

### 2.3. The Interface for Expert Annotation

To display the results and enable the experts to annotate TSs and FSs, we created a dedicated MATLAB^®^ (R2021b) interface (Figure 1), based on the viewer from the Fetal Heart Rate Morphological Analysis (FHRMA) toolbox [22]. The expert drew a rectangular window to precisely select the beginning and end of FS and TS periods. The interface could display an interpolated signal during a period of MS as a lighter line and thus could emphasize short FS periods which would only concern a few pixels on the screen and could easily have been overlooked. The interface shown in the figure is for annotation of the MHR and Doppler FHR at the same time. Another interface was dedicated to the analysis of the FHR on the scalp ECG channel, the period selection of which was independent of the Doppler/MHR channels.

The training datasets were analyzed simultaneously and consensually by one or two experts: an engineer and a senior obstetrician, both of whom performed research on and gave lectures on FHR analysis. The engineer’s involvement was important because it resulted in faster, more accurate use of the interface and provided a better understanding of (i) how an FS can arise and (ii) aspects of the recording that are important for machine learning.

The interface could also display a model’s results by using a color gradient for the FHR (TS = blue, FS = red) and the MHR (TS = violet, FS = cyan). Thus, the experts could annotate recordings while being aware of the mistakes made by the previously trained models. The interface is available as open-source code in the FHRMA MATLAB^®^ toolbox [22]. The FHRMA toolbox also included the model recoded in MATLAB^®^ because the training was performed with Python^®^ 3.7 and TensorFlow^®^ 2.4 (see Section 2.8.7).

### 2.4. Manufacturer-Specific Preprocessing

The CTG monitor measures the FHR via a Doppler sensor or a scalp ECG electrode and measures the MHR via an ECG sensor combined with the tocometer (TOCO+) on the belt or an oximeter on the finger. In all cases, the CTG monitor applies a proprietary auto-correlation algorithm to determine the heart rates from the raw signals. The autocorrelation algorithm’s time window depends on the sensor, thus creating different time lags. For Philips^®^ monitors, we have determined that the time lag between the scalp ECG FHR and the Doppler FHR is 1 s, the time lag between the Doppler FHR signal and the MHR measured with the tocometry belt is 5 s, and the time lag between the Doppler FHR signal and the MHR measured with the finger oximeter is 12.5 s.

We had to compensate for these time lags before comparing the heart rates. To estimate the size of the error generated by these time lags, we measured the mean difference in the absolutes value before and after compensation between (a) the signal measured during the MHR FS period with the Doppler FHR sensor and (b) the MHR measured with finger oximeter. For six recordings and a total of 60 min free of accelerations/decelerations, the mean difference was 5.0 bpm before correction and 3.5 bpm afterwards. Moreover, the difference was much greater when measured during accelerations or decelerations, which are often critical periods for FSs. Thus, this correction is very important.

Unfortunately, we did not record the type of MHR sensor in our database until May 2020. However, the type of sensor could be easily determined retrospectively because the MHR measured with the tocometry belt had a resolution of 0.25 bpm and that measured with the oximeter had a resolution of 1 bpm. Thus, periods with only whole numbers corresponded to an oximeter, and periods containing numbers with a decimal point corresponded to the ECG sensor combined with the tocometry belt.

A second preprocessing step (mostly for the scalp ECG electrode channel) concerned the fact that when a CTG monitor loses the signal, the previous FHR value is sometimes repeated for up to 30 s before the signal loss is displayed. We detected these “hold” periods as periods of more than 12 consecutive samples (each lasting 3 s) with the same value. We estimated that with this threshold, the average number of false detections of “holds” (periods during which the FHR coincidentally had exactly the same value over a period of 3 s) was <1 for 6 h of recordings - even when the FHR variability was low. These “holds” were replaced by MSs.

### 2.5. Preparation of the Input Matrix

We first normalized each signal (MHR or FHR) in beats per minute (bpm) by performing $\hat{HR}=\frac{HR-120}{60}$. Next, we coded MSs on an independent channel as $MS_{FHR}$ or $MS_{MHR}$: the absence of a signal was scored as 0 and the presence of a signal was scored as 1. During MSs, the corresponding $\hat{FHR}$ or $\hat{MHR}$ was set to 0.

The FSDop model’s input comprised the Doppler FHR signal and the MHR signal, whereas the FSHMHR and FSScalp model’s inputs comprised only the single, corresponding signal. Since there were no FSs for the MHR on the scalp ECG channels in our dataset, we did not input the MHR into the FSScalp model.

We added a channel to code the stage of delivery: for each sample, 0 corresponded to the first stage or antepartum, and 1 corresponded to the second stage. The second stage of delivery features more MHR accelerations, more FHR decelerations, more MSs, and many more FSs; hence, the delivery stage is important for adjusting the probabilities to the period.

Before DL can occur, the data have to be formatted in tables [batchsize×timesamples × channels]. Unfortunately, the duration differs from one recording to another. Although zero padding is the conventional way of handling differences in duration, our training dataset was mostly composed of short recordings (5 to 30 min) and a few longer signals (up to 4 h). Zero-padding short signals to 4 h would have created a high proportion (85%) of useless zero calculations, and cutting up the 4 h recordings into shorter parts would have prevented the models from detecting long-term features. We therefore grouped the short signals together into packages of approximately 4 h, and concatenated the signals inside a package to create a “unit”. A mini-batch was then composed of several units of the same duration (4 h). However, we had to tell the model to reset the internal status at each change of recording. To this end, we added first a reset sample (containing 0) between two recordings and then a “reset channel”, which was set to 0 most of time and to 1 on reset samples. The layers’ handling of this channel and these samples are described in Section 2.8.5 and Appendix B. Thus, Figure 2 shows the decomposed matrix obtained for a unit used as the input for FSDop.

For the FSDop dataset, the input matrix sizes were: 80×64800×6 (training dataset) and 20×64800×6 (validation dataset). These sizes enabled training on either TPU v3 (batch size: 40) or on Colab Pro^®^ with GPU T4 or P100 (batch size: 20). The batch size was limited to avoid memory saturation.

For FSScalp, the input sizes were 40×30600×4 for the training and validation datasets, since we worked with a batch size of 40 on either TPU v2 or on Colab Pro^®^ with GPU T4 or P100.

For FSMHR, the input sizes were 22×29000×4 for the training and validation datasets, since we worked with a batch size of 22 on Colab Pro with GPU T4 or P100.

Our initial trials used a constant window length of 30 min; shorter periods were padded with zeros. Unfortunately, the level of performance decreased when the method was applied to longer recordings; hence, we developed this recording concatenation system to train the models on longer periods. We decided to not subsample the recording because we did not know whether the models would be able to extract information from high-frequency signals; for example, a change in variability might suggest a switch between the FHR and the MHR.

### 2.6. Data Augmentation

Even though the number of recordings was high, we reasoned that DL could always benefit from more data. We augmented the data by applying a random transformation that changed a given recording into another realistic recording by:Removing the MHR channel completely (P = 10%).Adding periods of MS to the MHR or the FHR. We added a random number of MS periods of random duration (from 1 s to 10 min). On average, 15% of MS was added to the FHR signal and 25% of MS was added to the MHR signal; however; these percentage varied markedly from one recording to another.Transforming MHR TSs into MHR FSs by multiplying by 2 or dividing by 2 over a 1 min period, on average. We added 30 s (on average) of MS before and after each period. Overall, 12 periods of MHR FS per hour were added.Transforming FHR TS into FHR FS by multiplying or dividing it by 2 or by taking MHR ×2, ×3 or /2 and adding noise $\Delta_{MHR-FHR}$ (for FSDop only). The latter was created by taking periods from other recordings in which the Doppler FHR channel measured the MHR and the maternal sensor also measured the MHR. The noise was defined as the difference between the two signals. We then created 26 min of noise signal and added it randomly to the MHR, so that the FHR was not exactly equal to the MHR multiplied or divided by 2 and thus could be identified easily. MS could be added before or after the generated FS.Preserving the MHR channel from the possible changes listed above in 20% of cases and preserving the FHR channel in 15% of cases. Hence, neither the MHR nor the FHR were changed in 3% of recordings.Multiplying both the FHR and MHR by a random value, with a Gaussian distribution with an expected value of 1 and a standard deviation of 0.08. This multiplication was applied before normalization, so that the harmonics were still realistic.Cutting up the recording by adding a reset sample (Section 2.5) to the middle of recording (1 reset sample every 10 h, on average).

All the random parameters described above had strongly non-Gaussian distributions and very high variances. These parameters can be viewed in the source code in Python/TensorFlow^®^. Figure 3 shows a signal after random transformation.

### 2.7. Cost Function

For the FSMHR, FSDop and FSScalp models, the output for each time sample (at 4 Hz) is the probability of being an FS (rather than a TS). This is a binary classification problem, and so we used the conventional binary cross-entropy (CE) as the cost function. However, we also weighted the samples. The weighted CE (refered as $CE_{w}$) is defined in Equation (Equation 1) where $w_{i}$ is the weight of sample *i*, $C_{i}$ is the label of sample *i* (1 for FS, 0 for TS), $P_{i}$ is the model’s estimate of the probability of being an FS, and ln corresponds to the natural logarithm.
(1)$$CE_{w}=-\frac{\sum_{i,C_{i}=1}w_{i}ln\left(P_{i}\right)+\sum_{i,C_{i}=0}w_{i}ln\left(1-P_{i}\right)}{\sum_{i}w_{i}}$$

The weights were defined so that:Each sample not annotated by an expert (because either he/she was uncertain or the period did not contain difficult-to-interpret features) had a weight of 0. Thus, the cost function is not influenced by the model’s output (TS or FS) for these samples.Each MS sample had a weight of 0.For each specific period, the weight of annotated samples is set to$\sqrt{1/Ratio\;of\;annotated\;samples\;over\;the\;period}$. Thus, if a period is fully annotated, all the samples have a weight of 1. If a period of 25 min contains only 1 min of annotation, however, this annotated period is probably more important, and each annotated sample will have a weight of $\sqrt{25}$ (i.e., 5). The total weight of this period is then 5 times lower than that of an entirely annotated period but 5 times greater than that of a selection containing the annotated part only.

We considered that the two types of error (false positives and false negatives) were equally important and thus gave the annotated FS and annotated TS the same weights—even though the classes on the training dataset were highly unbalanced, as shown in Section 4.2. If, for example, a few samples are located in a deceleration trough, considering them as FSs would remove the deceleration from the recordings, and so fetal distress might be under-evaluated. In contrast, not removing an MHR period might cause a deceleration to be added to the recording, which would increase the likelihood of an unnecessary intervention.

### 2.8. The Model

#### 2.8.1. Using Bidirectional, Symmetric, Gated Recurrent Units (GRUs)

Due to the nature of the signal (with a variable length) and the need for a “synced many-to-many” model [23], we chose to use recurrent neural network (RNN) layers. An RNN layer calculates a state St for a time sample *t* by using both the input signals $I_{t}$ at *t* and the previous state $S_{t-1}$. Simple RNNs are fully connected: $S_{t}=f\left(W_{I}I_{t}+W_{S}S_{t-1}\right)$ where *f* is an activation function and $W_{I}$ and $W_{S}$ are the kernel and recurrent weight matrices, respectively. Simple RNNs are subject to the vanishing gradient problem and have difficulty retaining information for long periods. To solve the vanishing gradient problem, two other RNN architectures have been created: the long short-term memory (LSTM) [24] in 1997 and the GRU [25] in 2014. These architectures add a long-term memory using a gate system. Here, we chose to use a GRU because it requires slightly less weights to train, relative to an LSTM. To create a long-term memory, the GRU uses an update gate corresponding to a set of values of ]0,1[ for each state; 0 means that the state is updated independently of the previous value, and 1 means that the state is kept as it was at t−1. The update gate is a trainable layer. The GRU equations are given in Appendix B.

To optimally analyze a sample at a specific time, it is often necessary to look at what happened before the sample as well as what happened after. We therefore used bidirectional layers (i.e., a GRU applied by moving forward in time and a reverse GRU applied by moving backwards in time). We did not identify direction-specific features, and so the FS analysis would be the same in each direction; we therefore constrained the weights to be the same in each direction. Even though TensorFlow^®^ lacks a procedure for this, symmetric bidirectional RNNs can be effectively produced by concatenating the signal in reverse time order in the batch dimension. The RNN’s output for the reversed signals is then re-reversed and concatenated in the channel dimension.

#### 2.8.2. A Three-Layer GRU

We hypothesized that the first GRU layer could determine low-level features (such as the mean duration of continuous signal recording, the standard deviation of this duration, and the last FHR value from previous periods), that the second layer could determine medium-level features (e.g., the expected MHR value and the latter’s accuracy), and that the third and final layer could determine deeper features (e.g., the likelihood of whether the signal was the MHR, the MHR ×2, or the FHR ×2, etc.). We checked that the GRU would be capable of estimating these kinds of feature in a supervised way but did not check which features actually emerged in the model.

For accurate estimation, most of the features - even the deepest ones – might need the raw signal, and so we facilitate their transmission by concatenating them to the previous layer’s output. This idea is quite similar to the “shortcuts” used in the famous ResNet network [26] to jump over certain layers.

#### 2.8.3. Sparse Kernels

Even though the number of annotated outputs was high (≈3,000,000 binary values), most were obvious or were highly interdependent. Hence, to avoid overfitting, the number of trainable coefficients must be limited. However, we wanted to keep the numbers of states (e.g., the number of activations of each GRU) high enough to allow the emergence of all the required features. Hence, rather than reducing the number of states, we limited the number of possible interactions between states by setting zeros on recurrent and kernel matrices. For recurrent matrices, we set trainable values on n×n blocks in the diagonal and set the other matrix elements to 0. For kernel matrices, we set blocks of trainable values to connect some inputs to a small number of outputs (for details, see Section 2.8.6) and set the other matrix elements to 0. The total number of trainable coefficients was 349 for FSMHR, 7357 for FSDop (in additional to those independently trained with FSMHR), and 3445 for FSScalp.

The same computation could be performed by dividing GRU layers into small GRUs and concatenating them to avoid several useless multiplications by 0. However, when using a full-size matrix and adding the sparsity constraint, the operations were better parallelized, and the overall processes were faster.

#### 2.8.4. Dropout

Since the number of GRU states was low, strict dropout might erase some primordial features; we therefore preferred to use Gaussian dropout to add noise but keep the information. In our small number of trials, Gaussian dropout performed slightly better than a strict dropout, although the difference was not significant.

#### 2.8.5. Reset Constraints

Since several recordings can be concatenated into the same unit, we had to ensure that the state was reset when the GRU switched to another recording. Unfortunately, TensorFlow^®^ does not have a procedure to do this, and we could not find a procedure in the literature. By analyzing the GRU equations, we found that by adding a constraint to the kernel matrices and using the reset sample and reset channel (Section 2.5), we could force the GRU states to 0 (for details, see Appendix B).

#### 2.8.6. Overall Architecture

The three models’ respective architectures are shown in Figure 4 and Figure 5. The FSDop model requires prior computation of the FSMHR model because it is important to know whether the MHR sensor signal is true before comparing it with the FHR channel and thus estimating whether the Doppler FHR channel might be the MHR. We did try training FSDop and FSMHR at the same time but FSMHR overfitted faster than FSDop did. Hence, FSMHR was trained first, and the weights were set during FSDop’s training. FSScalp was independent of the other two models.

#### 2.8.7. Training

The models were initially developed in Python^®^ and TensorFlow^®^. They were trained using an Adam optimizer and a learning rate that fell progressively from 0.01 to 0.001 after 1000 epochs.

During the project’s development phase, approximately 15 different models/hyper-parameters were tested for FSMHR, 50 for FSDop and 15 for FSScalp; however, some of these models contained some minor bugs. For a given architecture and associated hyperparameters, performance and convergence speed varied greatly from one training session to another. For FSDop (the most complicated model), we selected (according to the validation data) the best of 16 training sessions with the same parameters. For the selected model, the minimum with validation data were reached after 45,000 epochs and then did not improve in the next 35,000 epochs. However, some models achieved almost the same results after approximately 3000 epochs and then stopped improving. The minimum was reached after approximately 5000 epochs for FSScalp and 10,000 for FSMHR.

For FSDop, the computation time was 40 s per epoch on a computer with an Nvidia^®^ RTX 2080 Ti GPU. Fortunately, we were able to access the Google^®^ TPU Research Cloud program. The computation time with TPU v3 was then 6 s per epoch. The long computation time was due to poor parallelization on the GRU because the parallelizable dimensions were small (number of states: 55; batch size: 40 × 2 for symmetry) and the non-parallelizable dimensions were large (sample number: 64,800 × 4 GRU layers × 2 batches per epoch).

For testing, we recoded the model in MATLAB^®^ and included it in the FHRMA toolbox [22]. We chose not to use the MATLAB^®^ Deep Learning toolbox to limit the user requirements. On a CPU (Intel^®^ i7-11800H), the computation time for FSDop for 1 h of recording was 0.9 s. When several recordings were computed in multiple threads, the computation time for 1 h of recording fell to 0.2 s.

## 3. Evaluation Methods

Performance was evaluated on data unseen by any model. Test datasets were created after the final model was established, so that we were not tempted to improve the models once we had seen the results with the test data (avoiding the risk of biased evaluation). Two evaluation systems and datasets were developed:(i)**A test dataset for routine clinical practice**: An evaluation of the three models on 30 recordings per model, selected at random and fully annotated by the experts. This evaluation was intended to provide an idea of the model’s performance with data obtained from routine clinical practice and that were not biased by selection criteria.(ii)**A dual-signal test dataset**: FSDop was evaluated on 34 recordings with the simultaneous scalp ECG sensor and Doppler sensor. The scalp ECG was used by two experts to set the ground truth. Two other experts analyzed the same periods but were blinded to the scalp ECG signal. This dataset was used to compare FSDop’s performance with that of the experts.

For each dataset and for each model, we measured the accuracy (setting a threshold of P = 0.5 for each classifier), the contingency table, the area under the receiver operating characteristic curve (AUC), and the CE. These metrics were also calculated for the training and validation datasets and for each stage (antepartum, the first stage of delivery, and the second stage of delivery). Given that a Doppler recording is not always accompanied by a simultaneous MHR recording in routine clinical practice (due to an obsolete CTG monitor, ill-trained staff or difficulties positioning the sensor), we also assessed the performance without considering the MHR channel.

The following subsection provides details of how the two test datasets were built.

### 3.1. The Test Dataset for Routine Clinical Practice

In the training and validation datasets, the proportion of FSs is higher than in the routine clinical practice since the periods were selected because they contained FSs or at least FS ambiguities. Thus, this test dataset was intended to assess the model’s performance without selection bias.

Thirty Doppler sensor recordings (containing the first stage of delivery and, unless cesarean section, the second stage) were selected at random from among those recorded in 2019. This dataset was used to evaluate FSMHR and FSDop.

Thirty scalp electrode recordings (during the first stage of delivery and, in some cases, the second stage) were selected at random from among those recorded between 1 January 2019, and 31 May 2021. This dataset was used to evaluate FSScalp.

The 60 recordings were analyzed by three experts (two obstetricians and a midwife, all of whom performed research on and gave lectures on FHR analysis). Each expert analyzed a third of the recordings. On the Doppler/MHR dataset, both the MHR and the FHR were fully annotated. Each sample of an FHR or an MHR from the start of the recording through to delivery was annotated as a TS, an FS, or an uncertain signal. On the scalp ECG dataset, the scalp ECG FHR channel was fully annotated as a TS, an FS, or an uncertain signal. The scalp FHR recording was generally shorter than the entire recording because the scalp electrode was applied as a second-line measure, after the Doppler sensor.

### 3.2. The Dual-Signal Test Dataset

The evaluation on test dataset for routine clinical practice is limited by potential errors made by the expert. Moreover, one cannot say whether the model is better or worse than the expert. Lastly, not all the recordings contained periods that the model might fail to analyze.

We therefore built a second test dataset by selecting all periods of a least 10 min between 1 January 2019, and 31 May 2021 (this inclusion period is after that of the training dataset A) on which both Doppler and scalp sensor signals were simultaneously recorded. This yielded 45 periods in 37 recordings.

Next, to form the solid ground truth, two experts (a senior obstetrician and an engineer) consensually annotated the Doppler channel with the help of the scalp ECG signal. These experts annotated all samples in a period as a TS, FS or uncertain signal, even when scalp ECG was temporarily missing. Next, two other experts (an obstetrician and a midwife) analyzed these periods but were blinded to the scalp ECG signals. The latter two experts had to annotate each sample in the period as a TS or an FS; annotating a signal as “uncertain” was not allowed. It was then possible to compare the method’s performance with that of the two experts.

## 4. Results and Discussion

### 4.1. Illustrative Results

Figure 6a shows an illustrative result for FSDop (and for FSMHR, even though there are no MHR FSs) over a period at the start of the second stage of delivery. Although the MHR was measured discontinuously, we can see that it contains accelerations and that these accelerations are synchronized with contractions (if recorded). In contrast, the FHR decelerates and is difficult to analyze because it crosses over the MHR curve. The start of the recording has a scalp ECG signal (in green), which shows the true FHR; the model did not see this scalp ECG signal. The probability of an FS estimated by the model is shown on a color scale. One can see that the method identified the period of FSs at minute 250 because it matches the MHR exactly. During a period in which the Doppler signal is superposed on the scalp ECG signal, the model predicted correctly that the signals were TSs. Another short period of FS appears to have been identified correctly at minute 256. Looking very closely, one can see that FSs at minutes 246 and 249 were not identified by the model; although this constitutes a minor error. This example would be very difficult to interpret in the absence of an MHR signal.

Figure 6b shows an illustrative result for FSScalp over the first stage of delivery. The few probable FSs appears to have been detected correctly.

Figure 6c shows another illustrative result for FSDop (and FSMHR, even though there are no MHR FSs) over a period corresponding to a first stage of delivery. The MHR showed accelerations and the FHR showed decelerations, as confirmed by the scalp electrode. The MHR and FHR values were sometimes identical. Although there was a long period of MS on the MHR channel, FSDop correctly identified the corresponding FHR FS—even during periods with MHR MS. The model might have failed to identify a possible short FS at minute 27.

Figure 6d shows a second illustrative result for FSDop and FSMHR over a period during which an epidural was given. There is a long period of MS on the Doppler channel. During this time, the MHR channel has FSs, most of which were detected. One can see that these MHR FSs do not prevent FSs on the Doppler FHR channel from being detected correctly.

Figure 7 gives an overview of the final interface within the FHRMA toolbox interface. It also shows the results of the morphological analysis (baseline, accelerations, decelerations, and UCs) using the weighted median filter baseline (WMFB) method [14]. The second half of this recording (second stage of delivery) is composed almost fully of FSs. Thus, by detecting FSs, the method did not flag up accelerations or decelerations during these periods. Lastly, the method highlighted the probably only true FHR signal (110 Hz), which might be a prolonged deceleration or borderline bradycardia.

### 4.2. Statistical Analysis

All the results for the training and test datasets are summarized in Figure 8. The left and right parts of the table correspond to the datasets and models, respectively. The most intuitive metric is accuracy, which was usually greater than 99%. However, since most of the signal samples are TSs, trivial classification of all samples as TSs would also produce a relatively high accuracy. This trivial model would have an accuracy corresponding to the “Percentage of “true” among annotated” column. An accuracy that is lower than the percentage of TSs means that the model removes more TSs than FSs. Hence, if we assume arbitrarily that a false negative has the same importance as a false positive (as in Section 2.7), the model would be of no use. This does not mean, however, that the method performs worse than chance, which will be measured with other metrics. Moreover, since FHR patterns are often redundant, we could also consider that removing FS is more important than not removing TS; a model might therefore still be of value.

The contingency table contains the sensitivity (Se), specificity (Sp), positive predictive value (PPV) and negative predictive value (NPV). Se+PPV>1 is equivalent to Acc>PercentageofTS, so this condition should be met for a useful model. However, the strongly imbalanced data meant that this is not a trivial problem, and so some models did not meet this condition. To perform better than chance, Se+Sp should be greater than 1; this was the case for all models.

The AUC is a guide to the classifier’s performance, independently of the threshold for the output probability. If the AUC > 0.5, we know that the model performs better than chance.

The CE is the most precise measure of performance (since it measures both accuracy and the model’s ability to recognize uncertainty) but is less intuitive for humans. A trivial random model in which all samples have a probability of P=Percentageof"truesignal" has CE=−P∗ln(P)−(1−P)ln(1−P). The CE is the metric optimized during the training (Section 2.7) but we did not weight the samples for the evaluation.

### 4.3. Results with the Test Dataset for Routine Clinical Practice

The accuracy values for this dataset showed that all three models are highly effective (FSDop: 99.66%, FSMHR: 99.92% and FSScalp: 99.93%). The percentage of TSs on the corresponding dataset were respectively 97.2%, 99.86% and 99.90%. The fact that the accuracy was greater than the percentage of TSs means that the models indeed rejected more FSs than TSs. This good performance is also confirmed by the respective AUCs (0.9992, 0.971 and 0.9792 for FSDop, FSMHR and FSScalp) and the respective CE (0.0112 (vs. 0.128 for the trivial model), 0.0044 (vs. 0.0108) and 0.0032 (vs. 0.0074)).

FSDop was the most useful model for routine clinical practice because the latter recordings contain many more FSs; the AUC of 0.9992 is impressive. The FSMHR and FSScalp models were less useful for this dataset, given to the very small number of FSs. However, the AUCs of FSMHR and FSScalp were respectively 0.971 and 0.978, which correspond to good, much-better-than-chance performance. Even though FSMHR and FSScalp have little effect (since there are very few FSs on these channels), it would make sense to implement them on the central monitor or as preprocessing steps.

FSDop’s performance fell to 97.88% for the second stage of delivery (Figure 8). This was expected because the second stage is often more complicated, with more MSs, more FSs, more FHR decelerations (which can be mistaken for the MHR), and more MHR accelerations (which can easily be confused with the FHR). The percentage of FSs in the second stage was 4.3%, although 12.8% of the second-stage samples were annotated as “uncertain” by the experts. This confirms that the second stage is more complicated to analyze, and the true proportion of FSs was probably around 10%—much more than the 2.8% in the first stage of delivery.

When we removed the MHR from the model’s input, the performance fell from 99.75% to 99.08% for the first stage of delivery, and from 97.88% to 94.72% for the second stage. Thus, the model is still effective for the first stage of delivery. However, for the second stage (and even though the AUC was 0.83), the model removed more TSs than FSs and so might not be relevant. For example, if these models were applied to the CTG-UHB public dataset [27], the absence of MHR data would limit their value. The difficulty of interpreting the second stage of delivery in the absence of an MHR signal was confirmed by the experts during their annotation of the study’s datasets; hence, this was not a method-specific problem. We encourage practitioners to check that the MHR is recorded well during the second stage of delivery, since poor obstetric decisions prompted by FHR/MHR confusion are probably more common than thought. We also suggest that the MHR could be recorded with a smartwatch, which might be more comfortable for the mother and possibly more reliable; we hope that CTG monitor manufacturers will study this possibility.

In this dataset, the experts were able to annotate features as “uncertain”, and so we expected very few incorrect (false) annotations. However, some of the models’ remaining errors might still be expert errors. The second test dataset was designed to overcome this problem but could be applied to FSDop only.

### 4.4. Results for the Dual-Signal Test Dataset

The dual-signal test dataset was more difficult to interpret than the routine clinical practice dataset. Nevertheless, the FSDop model’s accuracy was 97.29%, which was still higher than the proportion of true samples in the dataset (96.6%). The AUC was 0.965, which corresponds to good classification performance. The accuracy rates for the two experts (who analyzed the recordings under the same conditions as the models, i.e., blinded to the scalp ECG signal) were 96.91% and 97.29%; hence, the model was better than one expert and worse than the other. The degree of randomness in these statistics is difficult to assess. However, the statistics confirmed our impression that the method was as accurate as a competent practitioner although a well-trained expert who understands the FS mechanism described in Appendix A could probably do slightly better than the models. Thus, we encourage other researchers to try to improve the models’ performance levels; this one reason why we have shared the study’s resources. A few ideas for further research are given in Section 4.6.

### 4.5. Results with the Validation Dataset

As expected, performance on the validation dataset was slightly worse than performance on the training data. One cannot compare the levels of performance between the validation dataset and the test datasets because they do not have the same selection criteria. The validation dataset provided greater precision (but not greater accuracy), because there were more data in general and more FS data in particular. However, the results were probably influenced by evaluation bias, induced by our choice of the hyperparameters’ values in this dataset. The models’ accuracies were very satisfactory (99.30% for FSDop, 98.68% for FSMHR, and 99.90% for FSScalp) and much higher than the percentage of TSs (89.5%, 89.3%, and 99.4%, respectively); hence, the number of errors was divided by a factor of 15.0 for FSDop, 8.1 for FSMHR, and 6.0 for FSScalp. The FS rate was much lower in the FSScalp dataset (0.6%) than in the FSDop dataset (10.5%), and so FSDop is likely to be of use more frequently in practice. The FS rate for the MHR was high (10.7%) but this was mainly due to selection bias during annotation.

We did not consider it necessary to develop a test dataset for antepartum recordings because the latter do not present difficulties absent from the first stage of delivery. This was confirmed by the accuracy on the antepartum validation dataset (99.37%), which was approximately equivalent to the accuracy on the first stage of delivery (99.49%) for much the same percentage of TS (88.2% vs. 89.4%, respectively). The only details which did not work with the first models we developed were present in a recording with an FHR baseline above 210 bpm. The problem was solved by data augmentation by random multiplication (as described in Section 2.6).

Poor performance in the second stage (with a censored MHR channel) was apparent in the validation data. Indeed, the CE was 0.2619, which was not significantly better than the CE of a trivial random model (0.273). We tried to train another model independently for this particular task (i.e., the second stage of delivery without an MHR sensor) and achieved a CE of around 0.23; however, we considered that this performance was too poor to justify adding another model. The performance improves drastically when an MHR sensor is present, and it would be better to change current medical practice and ensure that the MHR is always recorded.

### 4.6. Perspectives

The models’ levels of performance were satisfactory but could probably be improved:Our models do not use the information from the tocography signal. There are some complicated cases in which the synchronization with contraction can help to determine whether the FHR is a deceleration or an FS. Adding this information to the model would require the automatic detection of the start and end of the UCs; this is unlikely to be an easy task and would probably not emerge accurately from optimization of the FS detection task in DL.Access to the raw Doppler signal from which the auto-correlation is computed (for estimation of the FHR) would provide additional information for recognition of the FS. However, obtaining the raw Doppler signal is not possible with most commercial devices.Some of our preprocessing steps (FHR/MHR delay compensation, and removing “holds”) were specific to Philips^®^ monitors and should be adapted for other manufacturers’ monitors. Once the delay is compensated for, we expect our models to work on other monitors but we have not yet evaluated this aspect.We have not yet determined whether FS detection can improve the performance of automatic FHR analysis for acidosis detection. Our initial results suggest that some of the computed features were improved by this preprocessing but that others were not (e.g., the deceleration surface). This is difficult to explain, and our future research will focus on this topic.Once trained, the models have a very short computation time (<1 s per hour of recording). However, the training is lengthy, and the differences in performance between two training sessions obliged us to train the models several times. To improve the models, we recommend reducing the training time by (for example) mixing convolutive and recurrent neural networks or using transformers. It would also be better to create a model with less variability between training sessions.Data augmentation enabled us to greatly improved the models. Our efforts to generate realistic FSs from TSs could probably be continued. Other techniques (jittering, elongation, warping, etc.) could be used. It might also be possible to synthesize realistic Doppler FHR signals from the scalp ECG signal, although we have not yet tried this.It might be possible to increase the number of recordings with a solid ground truth (i.e., better than the analysis by experts, who had to use the same information as the models) by putting two Doppler sensors (rather than one) on the belt. Although this would be less precise than a scalp ECG, adding the latter sensor with no medical justification is ethically problematic. Thousands of recordings could be recorded and annotated automatically by considering that two similar signals are necessarily either both TSs or (less likely) both FSs. If two signals differ, one is likely to be false. Originally, we tried this method for annotating training dataset A (with dual Doppler/Scalp ECG sensors). Although this method worked, only 90 recordings were concerned and it was simpler and more informative to use expert annotation with the scalp ECG as an indicator.For real-time applications, the computation time is not a problem. However, since we used a symmetric RNN, the whole signal is used for the analysis of each instant. When new signal is added at the end of the recording, the analysis of the beginning of the recording might change (albeit rarely). Let us take the example of a 120 Hz heart rate acquired on the FHR Doppler sensor and an MHR sensor that is incorrectly positioned at the beginning of a recording. The signal at 120 Hz is most probably the FHR, and so TS. Thirty minutes later, the MHR sensor has been repositioned and we can see a maternal tachycardia (also at 120 Hz). Thus, the method’s output might change and might indicate that the signal at the start of the recording is an FS. One can note that most experts would also change their mind after analyzing a recording of this type; hence, this is not a method-specific limitation. In clinical use, it is not possible to wait for even 5 min before flagging up an FS and sending an alarm: the sensor must be checked as soon as possible. Consequently, the display interface should be well thought out; it is essential to display the analysis immediately while perhaps indicating that the analysis is not definitive. If the model’s output changes significantly, the interface must display it clearly for reasons of medical traceability. We have not yet evaluated the frequency of change in the model’s output in function of the time to the added signals at the end of recording.

## 5. Conclusions

We developed DL models for detecting FSs in FHR and MHR recordings. The detection of FSs is particularly useful for Doppler recordings; the high probability of capturing the MHR instead of the FHR means that FSs are particularly frequent. The models performed at expert level, although a well-trained expert could probably do better in some cases. We hope that these models will be able to improve clinical care by alerting the practitioner to the possible presence of FSs in the FHR recording. Moreover, the models can be used to preprocess FHR recordings for subsequent automatic analysis. We showed that FS detection is a complicated problem and is often neglected in the literature; however, tackling this problem might have a major impact on both clinical care and automatic analysis.

Our results suggest that the MHR sensor is very important (and often essential) for recognizing the FS - particularly during the second stage of delivery. However, the MHR is often missing during the second stage of delivery. One simple, comfortable solution would be to add a smartwatch with a wireless connection. We also strongly encourage CTG monitor manufacturers to compensate for the delay between the FHR and the MHR (5 s or 12.5 s on a Philips system); this would make it easier to distinguish the MHR on the Doppler FHR channel, and this function could be patched (as display software) relatively easily into central monitoring stations.

The present work contributes to the field by (i) describing and formalizing the problem of automatic FS detection, (ii) offering a large, annotated dataset, and (iii) offering the first ever effective solution to the problem. Moreover, the approach presented here required several developments with a great degree of optimization (e.g., data augmentation, kernel/recurrent sparsity, and a symmetric GRU). We also developed a technique for concatenating recordings inside a GRU unit (Appendix B); this avoids useless operations due to zero padding, and might be useful for other problems.

To encourage other researchers to work on this problem, we have made all the data and our source code available via the open-access FHRMA project [22]. The test dataset annotations are not available on an open-access basis. Researchers who want to evaluate their models will have to send us their results for evaluation; hence, a competition has been opened. We also encourage researchers working on FHR signal processing to use our FS detection methods as preprocessing steps and to use our open-source WMFB method (also available in the FHRMA toolbox) [14] for FHR baseline estimation.

Our future research will evaluate the impact of FS detection on computation of FHR features for the detection of fetal acidosis.

## Figures and Tables

**Figure 1 biosensors-12-00691-f001:**
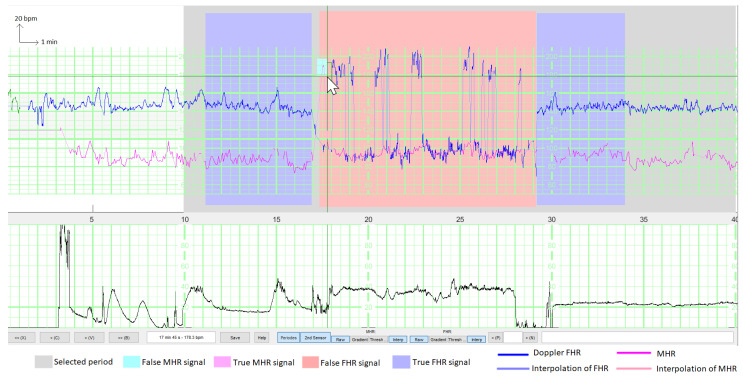
Illustration of the interface for annotating signals during a period with an epidural injection. Both the FHR (in blue) and the MHR (in purple) are shown, together with their interpolations (lighter colors) during MS periods (for better visualization of samples that are isolated from the rest of the signal). The user has selected a window for training the model (in grey) and has annotated two periods with a true FHR signal (in blue) and one period with a false FHR signal (in red). The false signals are either the MHR or the MHR × 2. Here, the user is selecting a few false MHR signals (in cyan) inside the rectangular box.

**Figure 2 biosensors-12-00691-f002:**
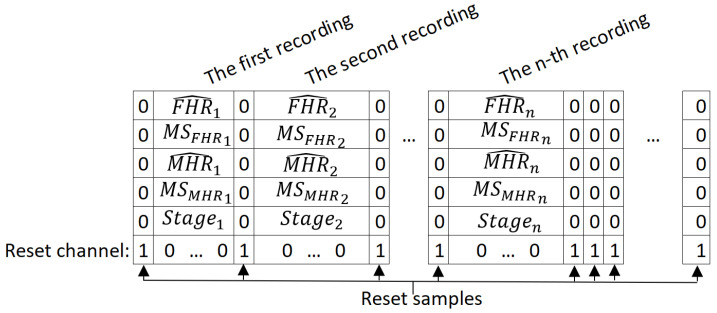
The input matrix for a unit in the FSDop model. $\hat{FHR}$ and $\hat{MHR}$ are the normalized FHR and the normalized MHR, respectively; $MS_{FHR}$ and $MS_{MHR}$ are respectively binary channels indicating whether the heart rate is an MS; Stage is a binary channel indicating whether the sample is in the first or second stage of delivery.

**Figure 3 biosensors-12-00691-f003:**
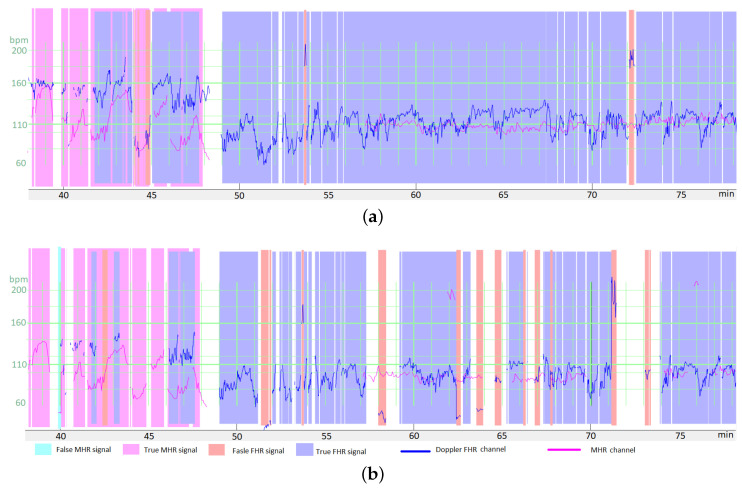
Example of the transformation of a period for data augmentation, with relatively high changes in the Doppler FHR channel (in blue) and the MHR channel (in purple). (**a**) The raw data in a period. (**b**) The period after random transformation. Expert annotations are shown as colored zones. Data augmentation consisted in adding MS and FS on both the FHR and MHR channels. Moreover, both the MHR and the FHR were multiplied by a λ≈0.9. The final signal corresponds to poor quality recordings but remains realistic. It should be noted that the MHR and FHR were not annotated by the experts for the entire recording (either because the experts were uncertain, or the period did not contain difficult-to-interpret features).

**Figure 4 biosensors-12-00691-f004:**
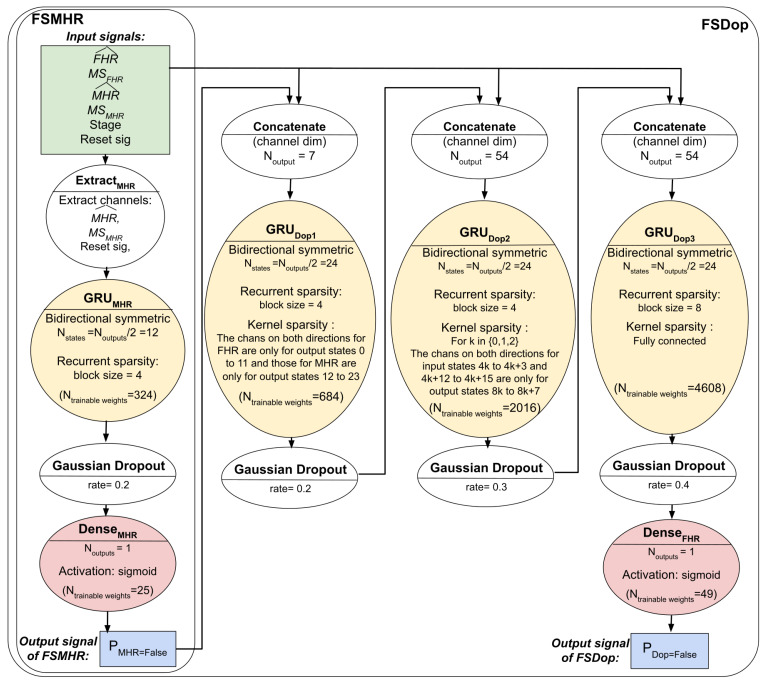
The FSMHR and FSDop models’ architectures and hyperparameters. For each MHR signal sample, FSMHR outputs the probability of being an FS and for each Doppler FHR signal sample, FSDop outputs the probability of being an FS.

**Figure 5 biosensors-12-00691-f005:**
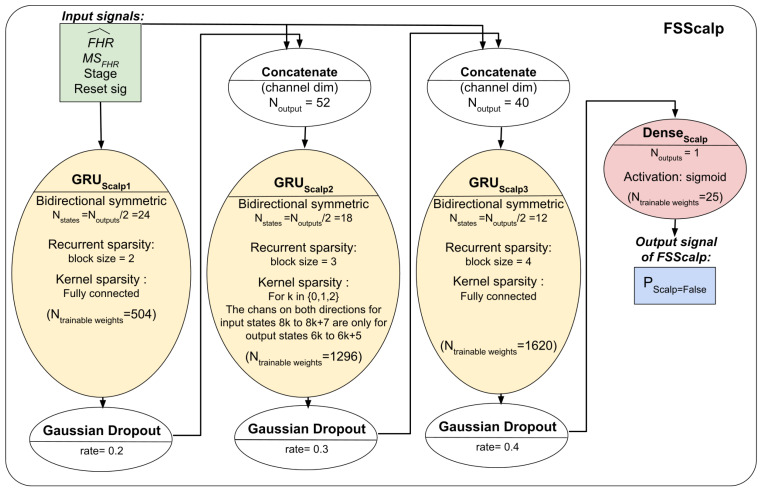
The FSScalp model’s architectures and hyperparameters. For each Scalp ECG FHR signal sample, FSScalp outputs the probability of being an FS.

**Figure 6 biosensors-12-00691-f006:**
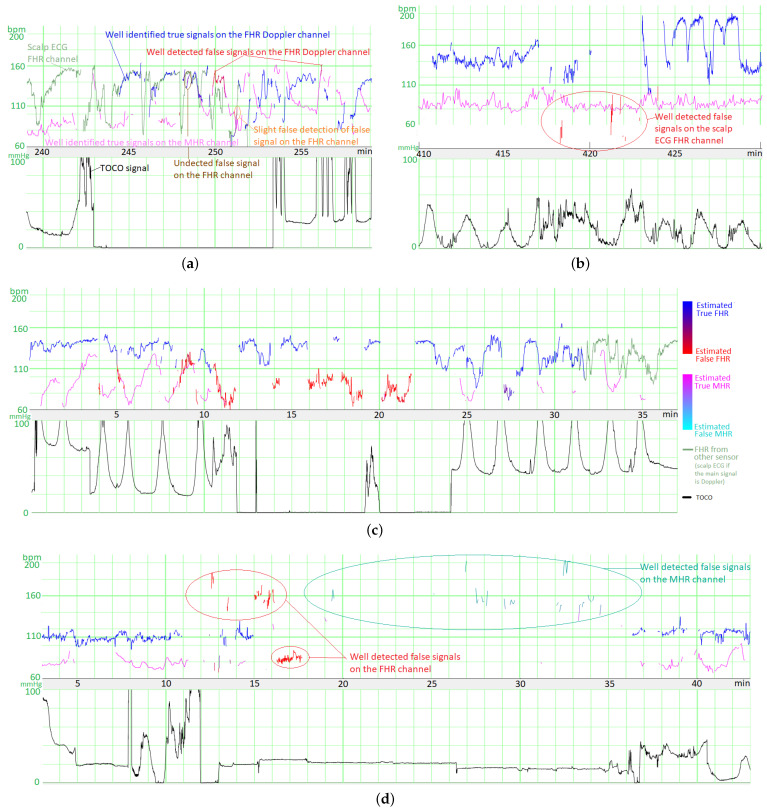
Examples of results for the three models (FSMHR, FSDop and FSScalp). (**a**) Results for FSScalp with a recording from the first stage of delivery. (**b**) Results for FSScalp with a recording from the first stage of delivery. (**c**) Results for FSDop and FSMHR with a recording from the first stage of delivery. (**d**) Results for FSDop and FSMHR with a recording from the first stage of delivery, during the administration of an epidural. The likelihood of an FS estimated by each model is represented as a color gradient. On examples (**a**,**c**,**d**), the blue/red signal corresponds to the Doppler channel and the green signal corresponds to the scalp ECG channel. On example (**b**), the blue/red signal is the scalp ECG signal. On all recordings, the time lag in the MHR had been corrected (as described in Section 2.4).

**Figure 7 biosensors-12-00691-f007:**
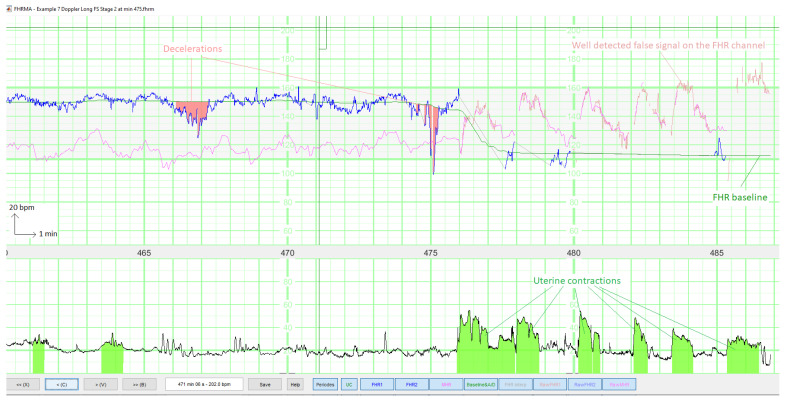
The FHRMA toolbox interface automatically displays both FSs and the results of a morphological analysis (baseline, accelerations, decelerations, and UCs). The FHR signal comes from the Doppler sensor. The second stage of delivery starts at 475 min.

**Figure 8 biosensors-12-00691-f008:**
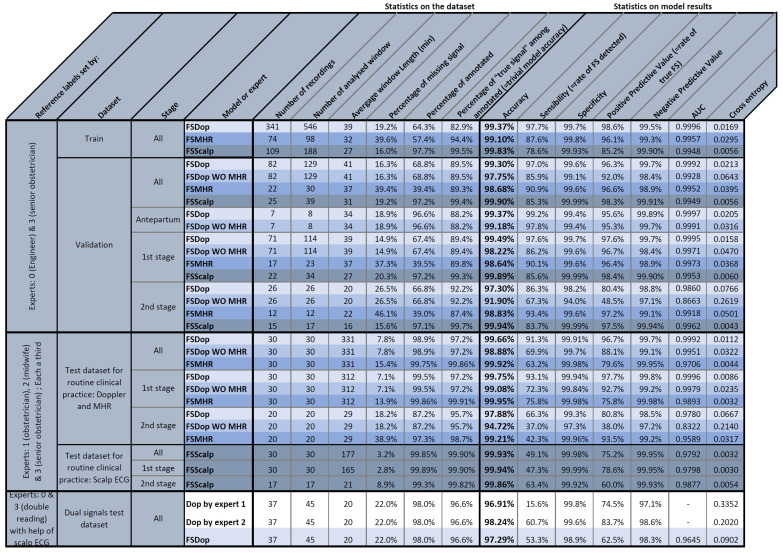
Statistical results for the study’s models and datasets.

## Data Availability

All data and source code are available in the Fetal Heart Rate Morphological Analysis Project at https://github.com/utsb-fmm/FHRMA (accessed on 10 August 2022).

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
