# Peer review of "Use of Deep Learning to Detect the Maternal Heart Rate and False Signals on Fetal Heart Rate Recordings"

_biosensors, 2022, doi:10.3390/bios12090691_

Round 1

Reviewer 1 Report

I have no comments. The paper is of high quality, models, data acquisition, data preparation, evaluation methods, and results are described in great detail.

I recommend the publication of this paper in its present form.

Author Response

Response to Reviewer 1 Comment

I have no comments. The paper is of high quality, models, data acquisition, data preparation, evaluation methods, and results are described in great detail.

I recommend the publication of this paper in its present form.

Thank you very much for that encouraging comment.

Reviewer 2 Report

This is an interesting paper presenting deep-learning model to predict false signals in FHR and MHR recordings and high potentials to be further utilized for alerting actual practitioners. Also, authors made the code in publicity, which encourages others to contribute and improve to this method. Here are some minor questions:

1) How did the window size selected? The work could benefit from showing performance when considering different window sizes.

2) Potentials of other types of conventional data augmentation methods for time-series such as jittering, elongation, and wrapping etc?

3) Authors aspect on using Transformer network rather than RNN & LSTM approaches to ensure long recorded signal patterns.

Reviewer 3 Report

-The paper should be interesting ;;;

-it is a good idea to add a block diagram of the proposed research (step by step);;;;;;

-it is a good idea to add more photos of measurements, sensors + arrows/labels what is what (If any);;;

-please add arrows to figures what is what;;;

-What is the result of the analysis?;;

-figures should have high quality (if there is such a possibility);;; 

-axes OX, OY/labels to figures should be added; - Figures 2, 5 please add SI units for example, Time [s], 

it can be done in a simple Paint program or power point;

-please add photos of the application of the proposed research, 2-3 photos (if any) ;;; 

-what will society have from the paper?;;

-Please compare the proposed method with other approaches/other methods/other neural network;;

-references should be from the web of science 2020-2022 (50% of all references, 30 references at least);;;

-Conclusion: point out what have you done;;;;

Round 2

Reviewer 3 Report

-Block diagram (well described) of the proposed research should be added;;; for example

input - signal XYZ

result - detection of what?

Author Response

-Block diagram (well described) of the proposed research should be added;;; for example

input - signal XYZ

result - detection of what?

Response: Thank you for this comment. We did not realize that inputs and outputs were not clear in our Figures. As recommended, we have updated the Figs. 3 and 4 as well as the graphical abstract to clarify that. We have also added in the legends

for fig. 3: "For each MHR signal sample, FSMHR outputs the probability of being a FS and for each Doppler FHR signal sample, FSDop outputs the probability of being a FS." 

for fig 4: "The FSScalp model’s architectures and hyperparameters. For each Scalp ECG FHR signal sample, FSScalp outputs the probability of being a FS. "